# Identifying MTHFD1 and LGALS4 as Potential Therapeutic Targets in Prostate Cancer Through Multi-Omics Mendelian Randomization Analysis

**DOI:** 10.3390/biomedicines13010185

**Published:** 2025-01-13

**Authors:** Huan Han, Hanwen Su, Zhihua Lv, Chengliang Zhu, Jingtao Huang

**Affiliations:** Department of Clinical Laboratory, Institute of Translational Medicine, Renmin Hospital of Wuhan University, Wuhan 430060, Hubei, China; hh940622@whu.edu.cn (H.H.); hanwensu@whu.edu.cn (H.S.); lzh2007@whu.edu.cn (Z.L.)

**Keywords:** prostate cancer, drug target, LGALS4, MTHFD1, Mendelian randomization

## Abstract

**Background:** Prostate cancer remains one of the leading causes of cancer-related mortality in men worldwide. The treatment of it is currently based on surgical removal, radiotherapy, and hormone therapy. It is crucial to improve therapeutic prospects for the diagnosis and treatment of prostate cancer via drug target screening. **Methods:** We integrated eQTL data from the eQTLGen Consortium and pQTL data from UK Biobank Proteome Plasma Proteins (UKB-PPP) and deCODE health datasets. MR analyses (SMR, heterogeneity in dependent instruments (HEIDI), IVW, Wald ratio, weighted median, and MR-Egger) were used to screen candidate genes associated with prostate adenocarcinoma (PRAD) risk. Candidate genes were further verified through TCGA-based gene expression profile, survival analysis, and immune microenvironment evaluations. TIDE analysis was utilized to investigate gene immunotherapy response. Single-cell RNA sequencing data from the GSE176031 dataset were used to investigate the gene expression patterns. The Drug Bank, Therapeutic Target Database and Drug Signatures Database were utilized to predict targeted drugs for candidate genes. **Results:** MTHFD1 and LGALS4 were identified as promising therapeutic targets for PRAD, with evidence provided at multi-omics levels. LGALS4 was predominantly expressed in malignant cells and was correlated with enhanced immune checkpoint pathways, increased TIDE scores, and immunotherapy resistance. In contrast, MTHFD1was expressed in both tumor and microenvironmental cells and was associated with poor survival. Drug target prediction suggested that there are no currently approved drugs specifically targeting MTHFD1 and LGALS4. **Conclusions:** Our study identified MTHFD1 and LGALS4 as potential preventive targets for PRAD. However, future experiments are warranted to assess the utility and effectiveness of these candidate proteins.

## 1. Introduction

Prostate cancer is one of the most common malignancies of men worldwide. In 2018, there were approximately 1.3 million new cases of PCa and 307,000 deaths [1]. Although progress has been made through early detection by means of prostate-specific antigen (PSA) screening and therapeutic interventions, around 10% of patients develop metastatic disease, leading to a higher chance of PCa-related death [2,3]. Current treatment options for metastatic prostate cancer (mPCa) include androgen deprivation therapy (ADT), prostatectomy, radiotherapy, and immunotherapy. ADT remains the primary therapeutic approach for mPCa. However, 30–50% of patients who receive ADT eventually develop castration-resistant prostate cancer (CRPC). Moreover, ADT is linked to an increased risk of cardiovascular disease [4,5]. Additional agents, such as docetaxel, abiraterone, apalutamide, and enzalutamide, have demonstrated significant clinical efficacy in improving metastasis survival. However, they are also associated with a number of unique side effects, limiting their clinical application in certain patients [6,7,8]. Sipuleucel-T remains the only FDA-approved immunotherapy for prostate cancer of any indication to date. However, Sipuleucel-T should not be used in patients with metastatic disease to sites other than bone and lymph nodes because patients with visceral disease were excluded from the Sipuleucel-T trial [8]. In addition, the heterogeneity of prostate cancer (PCa), characterized by diverse genetic, proteomic, and transcriptomic landscapes, plays a critical role in tumor development, and complicates the establishment of universally effective treatment strategies. This complexity underscores the imperative for personalized medicine approaches to effectively manage and treat PCa [9,10]. Understanding the genetic and proteomic alterations contributing to PCa susceptibility and therapeutic resistance is critical to discovering new biomarkers and developing targeted therapies, accordingly [11].

Genome-wide association studies (GWAS) have been instrumental in identifying genetic variants associated with an elevated risk of PCa. However, GWAS findings frequently pinpoint loci that are non-coding or reside within regions harboring multiple genes, thereby posing significant challenges in accurately identifying causal genes and elucidating their biological mechanisms [12]. To address these limitations, expression quantitative trait loci (eQTL) analyses have been employed to link genetic variants with gene expression levels, facilitating the identification of genes whose altered expression may contribute to disease etiology [13]. Additionally, protein quantitative trait loci (pQTL) analyses extend this framework by associating genetic variants with protein abundance, thereby providing a more direct assessment of the functional implications of genetic variation [14].

Mendelian randomization (MR) provides a powerful approach for establishing causal relationships between genetic variants and disease outcomes by using these variants as instrumental variables [15]. Unlike traditional observational studies, MR effectively reduces the influence of confounding factors and prevents reverse causation, thereby enhancing the reliability of causal inferences. By integrating eQTL and pQTL data with GWAS through MR and summary-based Mendelian randomization (SMR) analyses, we can more accurately identify causal genes with significant therapeutic potential [16]. This comprehensive strategy not only elucidates the genetic architecture of PCa but also highlights druggable targets that may be exploited for targeted therapeutic interventions.

The tumor microenvironment (TME) plays a pivotal role in the progression, metastasis, and therapeutic response of PCa [17]. Within the TME, immune cells, such as T lymphocytes, macrophages, and myeloid-derived suppressor cells, interact dynamically with cancer cells, influencing tumor growth and facilitating immune evasion [17]. Notably, immune checkpoints, including PD-1/PD-L1 and CTLA-4, have emerged as critical regulators of immune responses in cancer, presenting promising targets for immunotherapeutic interventions. These checkpoints not only facilitate immune escape mechanisms but also serve as promising targets for immunotherapy, with checkpoint inhibitors showing success in certain cancer types. Additionally, the TME encompasses non-immune components, including fibroblasts, endothelial cells, and extracellular matrix elements, which collectively contribute to tumor progression and resistance to therapy. A deeper understanding of the cellular and molecular interactions within the TME is essential for identifying novel therapeutic targets and improving treatment strategies for PCa [18].

In this study, we developed a comprehensive analytical framework to systematically identify and validate druggable genes associated with PCa. By integrating GWAS, eQTL, and pQTL data, we employed MR and SMR analyses to identify causal genes linked to PCa risk. Subsequent validation in independent datasets, combined with clinical expression and survival analyses using The Cancer Genome Atlas (TCGA), enabled us to assess the prognostic significance and immune-related associations of the identified genes. Additionally, we utilized single-cell RNA sequencing (scRNA-seq) data to elucidate the cellular context of gene expression within the TME and conducted drug prediction models and molecular docking simulations to explore the therapeutic potential of the top candidate genes. This integrative approach not only uncovers novel genetic determinants of PCa but also establishes a translational pathway toward the development of targeted therapies and precision medicine strategies.

## 2. Materials and Methods

### 2.1. Study Design

This study developed a comprehensive analytical framework, as shown in Figure 1, to systematically identify and validate druggable genes associated with PRAD. First, eQTL data from the eQTLGen Consortium were used as instrumental variables in the MR analysis to identify potential causal genes. This was followed by an SMR analysis and HEIDI tests that assessed the causal roles of these genes in PRAD.

These findings were further validated at a second stage using the pQTL data obtained from the UK Biobank Proteome Plasma Proteins (UKB-PPP) and deCODE health studies. Sensitivity analyses were stringently performed to ensure the robustness of the results.

To further investigate the clinical relevance, gene expression profiles, prognostic significance, and immune-related associations, an analysis using the TCGA database was performed. Colocalization analyses were conducted to assess whether the causal variants arose from shared or distinct genomic regions. Drug prediction models and molecular docking simulations were finally performed and gave useful insights into the therapeutic potential of identified targets and broader perspectives for the targeted treatments of PRAD.

### 2.2. Exposure Data

We curated the eQTL data from the eQTLGen Consortium (https://eqtlgen.org/ (accessed on 4 October 2024)) [19], including 16,989 cis-eQTL genes identified by a genetic analysis of 31,684 blood samples from healthy individuals of European ancestry. To study the genetic architecture of plasma protein abundance and its relation to disease, we integrated data from two large-scale GWAS: the UKB-PPP and the deCODE health study. The UKB-PPP used the Olink platform to analyze blood plasma from 54,219 UK Biobank participants and derived important genetic associations with the levels of 2923 proteins [20]. The deCODE health study similarly utilized the high-throughput SomaScan platform to quantify 4907 aptamers associated with plasma protein levels in a cohort of 35,559 Icelandic individuals [21].

For our analysis, we specifically selected index cis-pQTLs as instrumental variables, focusing on those that met genome-wide significance thresholds (*p* < 5 × 10^−^⁸). These were restricted to cis-pQTLs falling within 1 Mb of the gene encoding the respective protein. We estimated LD for our instrumental variables using data from the European panel of the 1000 Genomes Project. Moreover, colocalization analyses were performed only for proteins with index cis-pQTLs from the UKB-PPP dataset in order to further strengthen the robustness of our findings.

### 2.3. Outcome Data

The PRAD cohort dataset utilized in this study was obtained from FinnGen Release 11, which was made publicly available in June 2024 and can be accessed at https://www.finngen.fi/ (accessed on 4 October 2024) [22]. This dataset includes 17,258 cases of PCa and 143,624 control individuals.

### 2.4. SMR Analysis and HEIDI Test

SMR and HEIDI methods integrate GWAS and eQTL summary data to differentiate between pleiotropy and linkage [23]. In the SMR analysis, associations between genetic variation and gene expression were identified by the use of summary statistics from both; significant associations were defined at *p* < 0.05. The HEIDI test was further used to determine if the SMR association was due to either pleiotropy (a single variant affecting multiple traits), or linkage (closely linked variants affecting different traits). A significant HEIDI result (*p* < 0.05) indicates pleiotropy. Together, these methods provide strong evidence for causal links between PCa and gene expression.

### 2.5. Mendelian Randomization Analysis

In this study, the MR analysis was performed using the “TwoSampleMR (version 0.5.8)” R package. To ensure the validity and robustness of our instrumental variables (IVs), we followed a rigorous approach. First, we selected genetic variants with an F-statistic ≥10 to confirm their strength as instruments. We then prioritized variants with low linkage disequilibrium (LD; r^2^ < 0.1) based on the European 1000 Genomes reference panel, ensuring IV independence [24]. To avoid reverse causation, we applied Steiger filtering to exclude SNPs with greater variance in the outcome than the exposure [25].

For genes with multiple IVs, we used the following four complementary methods: inverse-variance weighted (IVW); Wald ratios; weighted median; and MR-Egger. IVW, chosen as the primary method due to its statistical power, assumes that all IVs are valid [26]. In addition, the Wald ratio was used to assess individual SNPs [27], while the weighted median method remained reliable even with up to 50% invalid instruments [28]. MR-Egger, which accounts for pleiotropy, included an intercept term to estimate average pleiotropic effects and the slope for causal effects [29].

We also used some statistical tests to further assess horizontal pleiotropy and heterogeneity. We also tested for heterogeneity using Cochran’s Q test and evaluated horizontal pleiotropy with MR-Egger regression intercepts and the MR-PRESSO method, which identifies directional pleiotropy [30].

### 2.6. Clinical Expression and Survival Analysis

Gene expression data and corresponding clinical information for PRAD were retrieved from the TCGA database (https://portal.gdc.cancer.gov (accessed on 4 October 2024)). Both count and TPM data were downloaded using R (version 4.3.0), with TPM values normalized via log2(TPM + 1) transformation. After excluding incomplete RNA sequencing and clinical data, 554 samples were included in the analysis.

Gene expression was assessed using the Kruskal–Wallis (KW) non-parametric test on count data. Kaplan–Meier (KM) survival analysis compared survival between two groups, with statistical significance determined by the log-rank test. KM curves, *p*-values, and hazard ratios (HR) with 95% confidence intervals (CI) were calculated using log-rank and univariate Cox regression. All analyses were conducted in R, with *p* < 0.05 considered significant.

### 2.7. Immune Scoring and Immune Checkpoint Analysis

To investigate the relationship between gene expression and the immune microenvironment in PRAD, RNA sequencing data were log2(TPM + 1) transformed. After excluding incomplete samples, 498 cases were analyzed and divided into high- and low-expression groups based on MTHFD1 and LGALS4 expression (249 cases each). Immune scores for each group were calculated using the ESTIMATE algorithm, which assesses stromal, immune, and total scores. The expression of immune checkpoint genes, such as ITPRIPL1, SIGLEC15, TIGIT, CD274, HAVCR2, PDCD1, CTLA4, LAG3, and PDCD1LG2, was compared between the groups. Statistical analyses were performed in R, with differences evaluated using the Wilcoxon rank-sum test (*p* < 0.05 considered significant).

### 2.8. Analysis of MTHFD1 and LGALS4 Expression in PRAD

Single-cell RNA sequencing data from the GEO database (GSE176031) were used to investigate the transcriptional profiles of MTHFD1 and LGALS4 in PRAD. scRNA-seq data, in .h5 format, together with annotation files, were obtained from the TISCH database (http://tisch.comp-genomics.org/ (accessed on 4 October 2024)). Data were processed with MAESTRO and Seurat for quality control, normalization, scaling, and clustering. t-SNE (t-distributed stochastic neighbor embedding) re-clustering identified cell types, which were annotated using marker genes. The expressions of MTHFD1 and LGALS4 were evaluated in tumor, stromal, and immune cells, and were represented using violin plots and heatmaps.

### 2.9. Drug Target Prediction

Potential drugs targeting MTHFD1 and LGALS4 were identified using two drug target prediction platforms: DrugBank (https://go.drugbank.com/ (accessed on 5 October 2024)) and the Therapeutic Target Database (TTD, https://db.idrblab.net/ttd/ (accessed on 5 October 2024)). These resources provided comprehensive details, including approved drugs, agents in clinical trials, and experimental compounds, along with their drug IDs, mechanisms of action, and current development stages.

Meanwhile, MTHFD1 and LGALS4 were submitted to the Drug Signatures Database (DSigDB, http://dsigdb.tanlab.org/DSigDBv1.0/ (accessed on 5 October 2024)) to identify potential protein–drug interactions. DSigDB contains 22,527 gene sets and 17,389 compounds linked to 19,531 genes, aiding the discovery of connections between drugs, chemicals, and target genes [31]. MTHFD1 and LGALS4 were also uploaded to the Enrichr tool (https://maayanlab.cloud/Enrichr/ (accessed on 5 October 2024)) to predict potential drug candidates targeting these genes [32].

## 3. Results

### 3.1. Discovery of Potential cis-eQTL Genes and PRAD

We integrated data from the eQTLGen Consortium, UKB-PPP, and deCODE datasets to identify genes significant at both RNA and protein levels. In the discovery phase, we identified 16,989 cis-eQTL genes from eQTLGen, validated by 1974 cis-pQTL proteins from UKB-PPP, and 1703 from deCODE.

We found the following gene overlaps between the datasets: 851 genes between eQTLGen and UKB-PPP; 654 genes between eQTLGen and deCODE; 241 genes between UKB-PPP and deCODE; and 644 genes overlapping across the three datasets of SMR, UKB-PPP, and deCODE.

For PRAD patients (17,258 cases, 143,624 controls from FinnGen), SMR and HEIDI analyses identified 1177 cis-eQTL genes associated with PRAD risk that passed the SMR test criteria (*p* < 0.05) and the HEIDI test criteria (*p* > 0.05) (Figure 1, Appendix A).

### 3.2. MR Analysis of pQTLs in Validation Phase

To examine the causal effects of candidate genes on PRAD at the protein level, we used cis-pQTL protein data from the UKB-PPP and deCODE databases. Because of missing gene data, only 26 genes from the discovery phase were included in the MR analysis.

In the validation phase, we found that C1QA, MARCO, TXNDC15, IGFBP3, TNFRSF10C, TRIM5, LAYN, JAM3, RNF43, PPP1R14A, SPINT2, LGALS4, TOR1AIP1, AGER, CILP, CEACAM21, and IL10RB were significantly associated with PRAD risk. MARCO, LAYN, and IL10RB were linked to reduced PRAD risk, while C1QA, TXNDC15, IGFBP3, TNFRSF10C, TRIM5, JAM3, RNF43, PPP1R14A, SPINT2, and LGALS4 were associated with increased risk (Figure 2, Appendix A).

The direction of effect for these 13 proteins was consistent with the SMR results, showing no evidence of heterogeneity or pleiotropy (Appendix A). Steiger filtering also confirmed the causal relationship (Appendix A). Four proteins—TOR1AIP1, AGER, CILP, and CEACAM21—were excluded due to inconsistent effects between the discovery and replication phases.

The deCODE database analysis identified PLXND1 and TNFSF14 as genes linked to a reduced risk of PRAD (Figure 3, Appendix A), while NEGR1, REG4, ECM1, GNMT, POR, MTHFD1, and GSTZ1 were associated with increased PRAD risk. The effects of PLXND1, TNFSF14, ECM1, GNMT, POR, MTHFD1, and GSTZ1 were consistent with the SMR results, showing no heterogeneity or pleiotropy in both proteins across four analytical methods (Appendix A). However, NEGR1 and REG4 were excluded due to inconsistencies between the discovery and replication phases.

Then, using cis-pQTL data in UKB-PPP and deCODE, the MR analysis identified CRHBP, GFRAL, NTRK3, SERPINA1, SERPINA3, and SLAMF7 as being associated with increased PRAD risk, whereas COL2A1, MSMB, POSTN, and RNASE3 were associated with reduced PRAD risk (Figure 4, Appendix A).

### 3.3. Clinical Expression and Survival Analysis

Using RNA sequencing data from TCGA for PRAD, we validated the previously identified candidate drug-associated genes for PRAD. Figure 5 shows the significant differential expression between the tumor and paired-normal tissues. Specifically, RNF43 (*p* = 2.9 × 10^−4^), IL10RB (*p* = 1.4 × 10^−3^), GSTZ1 (*p* = 3.0 × 10^−5^), COL2A1 (*p* = 0.011), SERPINA3 (*p* = 2.6 × 10^−3^),SPINT2 (*p* = 0.035), TRIM (*p* = 0.035), GNMT (*p* = 0.037) and MTHFD1 (*p* = 2.7 × 10^−6^) were upregulated in tumor samples compared to their normal counterparts, while IGFBP3 (*p* = 1.2 × 10^−3^), TNFRSF10C (*p* = 4.3 × 10^−5^), LAYN (*p* = 7.3 × 10^−8^), JAM3 (*p* = 4.7 × 10^−7^), PPP1R14A (*p* = 1.4 × 10^−9^), LGALS4 (*p* = 5.8 × 10^−6^), ECM1 (*p* = 2.6 × 10^−3^), CRHBP (*p* = 8.1 × 10^−5^), MSMB (*p* = 0.011), NTRK3 (*p* = 2.3 × 10^−3^) and RNASE3 (*p* = 6.6 × 10^−5^) were downregulated.

Due to low mortality, only a few genes were linked to overall survival (OS). TNFRSF10C (HR (95% CI) = 0.19 (0.055–0.66), *p* = 3.91 × 10^−3^) and CRHBP (HR (95% CI) = 0.24 (0.067–0.85), *p* = 1.64 × 10^−2^) were linked to good OS, while LGALS4 (HR (95% CI) = 5.56 (1.20–25.86), *p* = 1.40 × 10^−2^), IL10RB (HR (95% CI) = 5.94 (1.28–27.57), *p* = 9.84 × 10^−3^), MTHFD1 (HR (95% CI) = 6.55 (1.76–24.45), *p* = 1.26 × 10^−3^), and POSTN (HR (95% CI) = 3.85 (1.00–14.76), *p* = 3.52 × 10^−2^) were associated with poor overall survival (Figure 6).

Thus, based on the survival analysis of the TCGA database, all genes except MTHFD1 and LGALS4 were excluded due to inconsistencies in the directionality of their effects.

### 3.4. Potential Immune Therapy Response Prediction

The TIDE (tumor immune dysfunction and exclusion) analysis revealed a strong association of MTHFD1 and LGALS4 expression with immunotherapy outcomes in PRAD. Specifically, higher LGALS4 expression was associated with higher TIDE scores (Figure 7A, *p* < 0.001), reflecting higher immune dysfunction and exclusion, and thus a poorer response to immune checkpoint therapies. In contrast, high expression of MTHFD1 was associated with lower TIDE scores (Figure 7B, *p* < 0.01), suggesting a different role in the tumor microenvironment. These findings highlight LGALS4 as a potential biomarker of poor immunotherapy response in PRAD.

Moreover, high LGALS4 expression was accompanied by increased immune checkpoint genes such as CD274 (PD-L1), CTLA4, HAVCR2 (TIM-3), ITPRIPL1, LAG3, PDCD1 (PD-1), PDCD1LG2 (PD-L2), SIGLEC15, and TIGIT (Figure 7C, *p* < 0.01 for all). MTHFD1 expression was related to high levels of PD-L2 and SIGLEC15 (Figure 7D). The upregulation of these immune checkpoint pathways may thus be indicative of the creation of an immunosuppressive tumor microenvironment that can foster mechanisms of immune evasion.

The elevated expression of LGALS4 therefore impairs immunotherapy response and facilitates the immune escape of PRAD by enhancing the signals of immune checkpoints.

### 3.5. Expression Patterns of MTHFD1 and LGALS4 in PRAD

Single-cell RNA sequencing (scRNA-seq) data from the GSE176031 dataset further illuminated these distinct expression patterns within the tumor microenvironment. LGALS4 demonstrates significantly elevated expression levels in malignant cells compared to other cell populations such as fibroblasts, T regulatory (Treg) cells, and epithelial cells (Figure 8A). The preferential expression in malignant cells suggests that LGALS4 may play a critical role in tumorigenesis and cancer cell-specific metabolic or signaling pathways. In contrast, the expression of MTHFD1 is more heterogeneous, with notable enrichment in mast cells, malignant cells, and macrophage/monocyte populations. This suggests the multifaceted role of MTHFD1 in cancer cell metabolism and the tumor microenvironment (Figure 8B).

Together, the distinct expression patterns of LGALS4 and MTHFD1 highlight their potential as biomarkers or therapeutic targets in PRAD. LGALS4’s association with malignant cells may reflect its role in tumor-specific pathways, while MTHFD1’s broader expression suggests its involvement in both cancer cell proliferation and microenvironmental interactions.

### 3.6. Drug Target Prediction of MTHFD1 and LGALS4

An analysis of the DrugBank and TTD (Therapeutic Target Database) databases revealed that there are no currently approved drugs specifically targeting MTHFD1 and LGALS4. Meanwhile, using the DSigDB drug database on Enrichr, etoposide, 1-(5-deoxypentofuranosyl)-5-fluoropyrimidine-2,4(1h,3h)-dione, capecitabine, and lactose are the candida drugs associated with LGALS4. The drugs associated with MTHFD1 are choline, choline hydroxide, dl-methionine, folic acid, methotrexate, dihydroergocristine, zinc sulfate, and alprostadil (Table 1).

## 4. Discussion

In the past decade, androgen deprivation therapy (ADT) remains the gold standard treatment for metastatic prostate cancer [33]. However, the majority of patients eventually develop castration-resistant prostate cancer (CRPC) [34]. With the advancement of molecular sequencing techniques, the emergence of biomarker-driven treatments offers a novel option for precision medicine for prostate cancer [35]. Our study suggested a significant correlation between the upregulation of MTHFD1 and LGALS4 and an increased risk of prostate cancer. Accordingly, the results emphasize the possibility of MTHFD1 and LGALS4 inhibitors as therapeutic interventions for mitigating PRAD risk.

Galectins are a family of soluble proteins that are involved in many physiological functions such as immune responses and cell migration, autophagy, and signaling [36,37,38]. LGALS4, a member of galectins, contains two carbohydrate recognition domains within the same peptide chain, with a high affinity for β-galactoside residues [39].

Previous studies have demonstrated that the expression of galectin-4 varies across different types and stages of cancer. The levels of galectin-2, -4, and -8 in blood were significantly elevated in patients with colon and breast cancers, particularly in those with metastases [40,41]. Tsai et al. demonstrated that LGALS4 was increased in clinical samples of prostate cancer and was correlated with tumor progression, poor survival, and cancer recurrence. Western blotting further revealed that LGALS4 activated the expression of pERK, pAkt, fibronectin, and Twist1, and lowered the expression of E-cadherin [42]. These results suggest that LGALS4 may play an important role in driving tumorigenesis and metastasis through the process of epithelial–mesenchymal transition (EMT). Moreover, LGALS4 was shown to induce apoptosis in T cells by binding CD3ε/δ and driving immune evasion in pancreatic cancer [43]. Notably, as a natural ligand for galactoglucan lectin-4, MUC1 induced EMT at the post-transcriptional level by modulating the expression of miRNAs in pancreatic cancer. Hypo-glycosylated forms of MUC1 have been shown to accumulate within cells, thereby initiating tumorigenic processes [44,45]. In summary, the available evidence supports the notion that LGALS4 is a potential therapeutic target in tumorigenesis and cancer progression.

Our results indicate that LGALS4 is predominantly expressed in malignant cells and is correlated with immunotherapy resistance and enhanced immune checkpoint pathways. Immune checkpoint inhibitors (ICIs) are monoclonal antibodies against immune checkpoint molecules that have shown remarkable efficacy in treating patients with a variety of cancers [46]. It has been demonstrated that CTLA4 and PD-1, along with its ligand PD-L1, have demonstrated significant therapeutic efficacy in cancers, including non-small cell lung cancer, metastatic melanoma, and bladder cancer [47]. Moreover, our results suggest that LGALS4 may be involved in immune evasion, a mechanism that contributes to resistance to both immunotherapy and androgen deprivation therapy. Therefore, targeting LGALS4 could improve the efficacy of ICIs and ATD in prostate cancer, especially in metastatic or castration-resistant cases.

MTHFD1 has three distinct enzymatic activities (5,10-methylenetetrahydrofolate dehydrogenase, 5,10-methenyltetrahydrofolate cyclohydrolase, 10-formyltetrahydrofolate synthetase), which played a significant role in the development of different types of cancers [48,49,50,51]. Research suggests that the overexpression of MTHFD1 in hepatocellular carcinoma was associated with poorer survival and recurrence [52]. The published data confirmed that the knockdown of MTHFD1 inhibited the proliferation, migration, and induced apoptosis of neuroblastoma (NB) cells. In addition, the study also suggested that MTHFD1 was upregulated in MYCN-amplified NB and correlated with the poor prognosis of NB patients [53]. In colorectal cancer (CRC), MTHFD1 regulated autophagic processes to facilitate tumor growth and metastasis through the PI3K–AKT–mTOR signaling pathway [54]. Furthermore, the expression level of MTHFD1 was associated with survival time, tumor size, TNM stage, histologic grade, and metastasis in non-small cell lung cancer and that MTHFD1 expression in tumor tissues and cells was significantly elevated compared to adjacent normal tissues and cells [55]. A current study found that MTHFD1 levels were not only associated with prognosis after CRT treatment of small cell lung cancer patients, but could also discriminate SCLC from both lung squamous cell carcinoma (LUSC) and lung adenocarcinoma (LUAD) [56]. Additionally, MTHFD1 may form a nuclear complex with the phosphatase and tensin homologue deleted on chromosome 10 (PENT) to assist dTMP synthesis in human prostate cancer cell lines that may have implications for targeting nuclear-excluded PTEN prostate cancer cells with antifolate drugs [57]. These findings emphasize the possibility of MTHFD1 as a potential therapeutic target for PRAD.

Our research has shown that MTHFD1 is widely expressed in malignant prostate cancer cells, mast cells, and macrophages. MTHFD1 is a key enzyme in single-carbon metabolism, involved in crucial processes such as methylation, nucleotide synthesis and amino acid metabolism [58,59,60]. Therefore, MTHFD1 inhibition may increase the sensitivity of PCa cells to chemotherapeutic agents, such as docetaxel, by disrupting the nucleotide and amino acid pathways essential for tumor growth. In addition, mast cells and macrophages play a key role in immune escape from tumors; MTHFD1 may alter the immune response in the tumor microenvironment by modulating the metabolism of these immune cells. In combination with ICIs, MTHFD1 inhibitors may help to restore immune cell recognition of, and attack on, tumors, thereby improving the efficacy of immunotherapy.

In summary, the targeting of LGALS4 or MTHFD, in combination with ADT, chemotherapy, and immunotherapy, could offer a novel approach to treating prostate cancer, particularly in cases where resistance to current therapies limits clinical efficacy.

There are numerous strengths in this study. Sensitivity analyses and reverse Mendelian randomization were utilized to confirm the robustness and reliability of the results. SMR and HEIDI tests were conducted for the elimination of horizontal pleiotropy. The TCGA database was used for further validation. Our study also has some limitations. First, MR analysis offers information on potential causal correlations, but its assumptions might not be entirely consistent with actual clinical trial conditions. Second, the primary data were obtained from European populations, which may hinder the extrapolation of our findings to diverse populations.

## 5. Conclusions

Our study identified MTHFD1 and LGALS4 as potential preventive targets for PRAD. LGALS4 may play a critical role in tumor-specific pathways, while MTHFD1 may be involved in both cancer cell proliferation and microenvironmental interactions. However, clinical trials are critical to ultimately assess the efficacy and safety of potential drug targets.

## Figures and Tables

**Figure 1 biomedicines-13-00185-f001:**
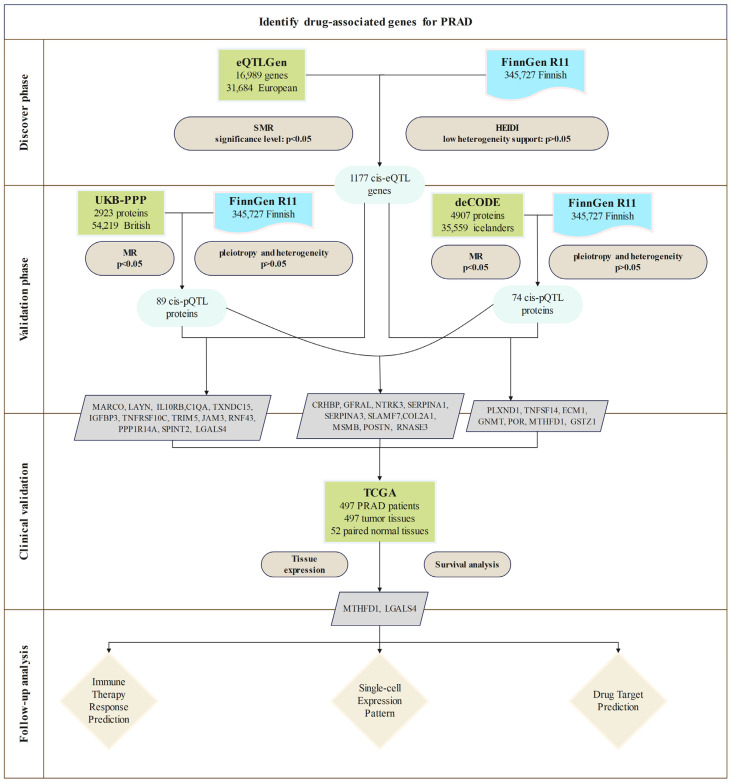
Overview of the study design.

**Figure 2 biomedicines-13-00185-f002:**
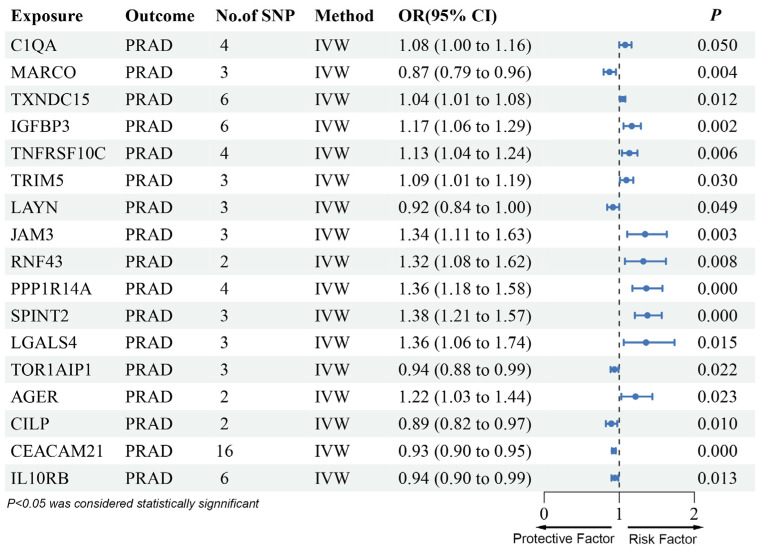
Forest plots illustrating the results of the validation stage for 17 essential genes in UK Biobank Proteome Plasma Proteins (UKB-PPP). OR: odds ratio, 95% CI: 95% confidence intervals; PRAD: prostate adenocarcinoma, SNP: single-nucleotide polymorphisms, IVW: inverse-variance weighted.

**Figure 3 biomedicines-13-00185-f003:**
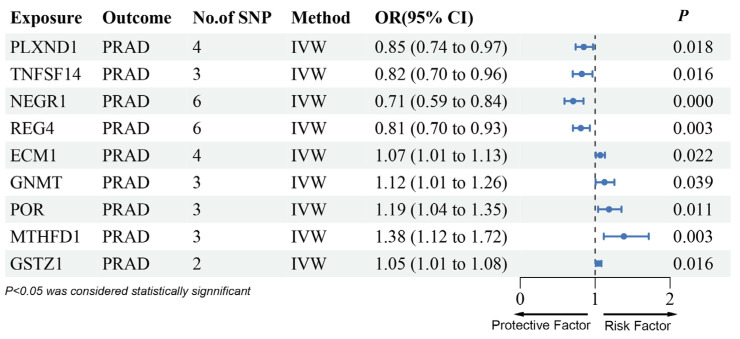
Forest plots illustrating the results of the validation stage for nine essential genes in deCODE.

**Figure 4 biomedicines-13-00185-f004:**
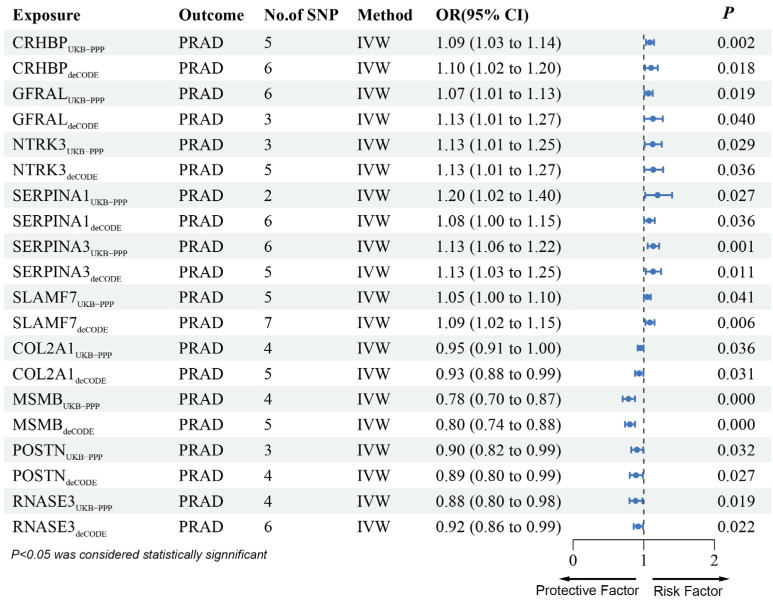
Forest plots illustrating the results of the validation stage for 10 essential genes in both UKB-PPP and deCODE.

**Figure 5 biomedicines-13-00185-f005:**
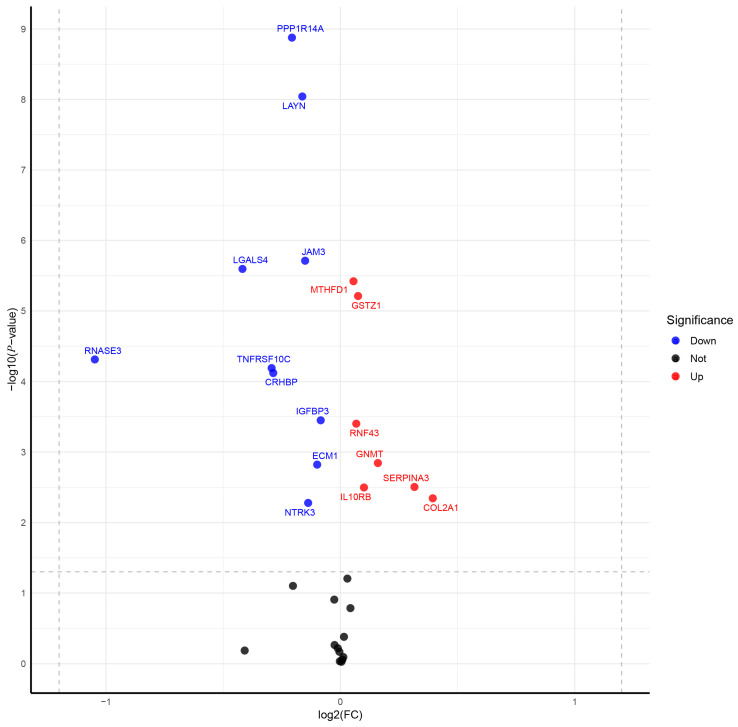
The expression of 30 essential genes between PRAD and paired-normal tissues. The volcano plot illustrates the differential gene expression analysis of 52 paired prostate cancer tissues compared to adjacent non-cancerous tissues. The x-axis represents the log2 fold change (log2FC) in gene expression, where positive values indicate upregulated genes in cancer tissues, and negative values indicate downregulated genes. The y-axis shows the −log10 (*p*-value), indicating the statistical significance of the differential expression. Significant genes are highlighted based on their fold change and *p*-value thresholds. Genes upregulated in prostate cancer are marked in red, while downregulated genes are marked in blue. Non-significant genes are displayed in black.

**Figure 6 biomedicines-13-00185-f006:**
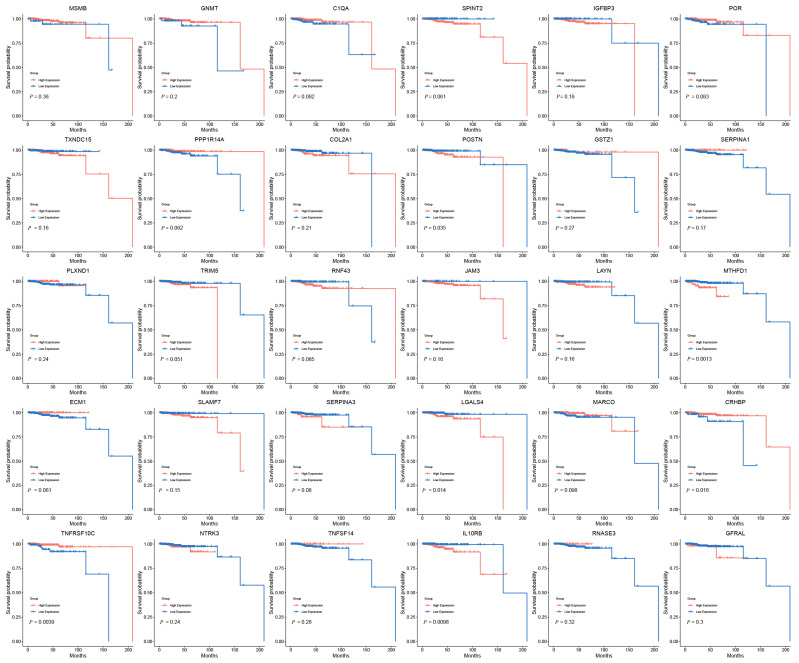
The survival analysis of 30 essential genes in PRAD patients. The KM survival curve distribution of different sample groups in the TCGA dataset, with log-rank tests conducted between groups.

**Figure 7 biomedicines-13-00185-f007:**
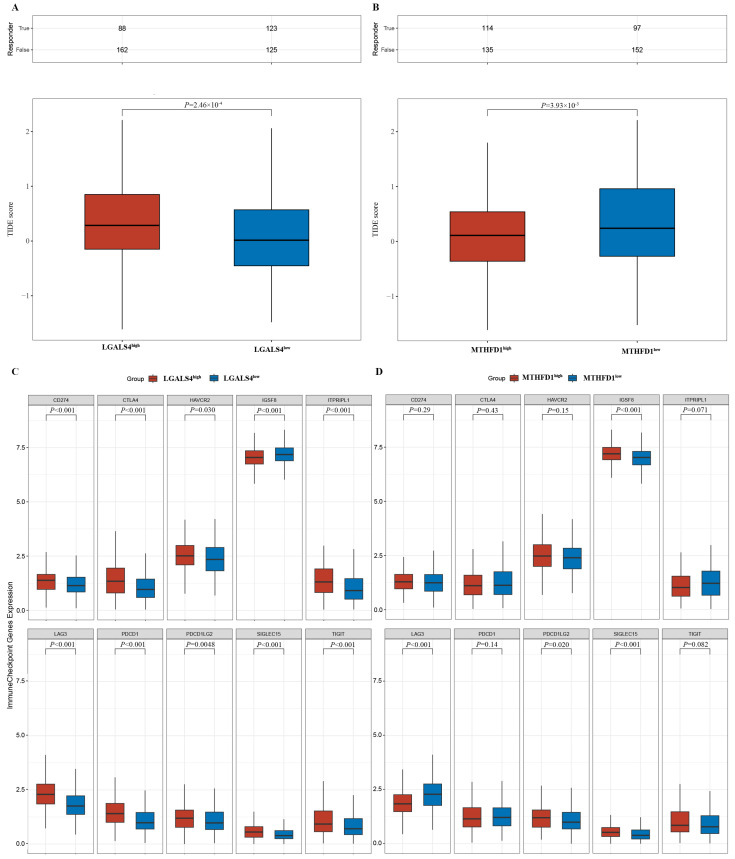
The potential immune therapy response prediction of LGALS4 and MTHFD1: (**A**) TIDE scores of immune response in LGALS4-high and LGALS4-low groups in the prediction results; (**B**) TIDE scores of immune response in MTHFD1-high and MTHFD1-low groups in the prediction results; (**C**) the distribution of immune checkpoint gene expression in LGALS4-high and LGALS4-low groups; and (**D**) the distribution of immune checkpoint gene expression in MTHFD1-high and MTHFD1-low groups. The significance was tested with the Wilcoxon test.

**Figure 8 biomedicines-13-00185-f008:**
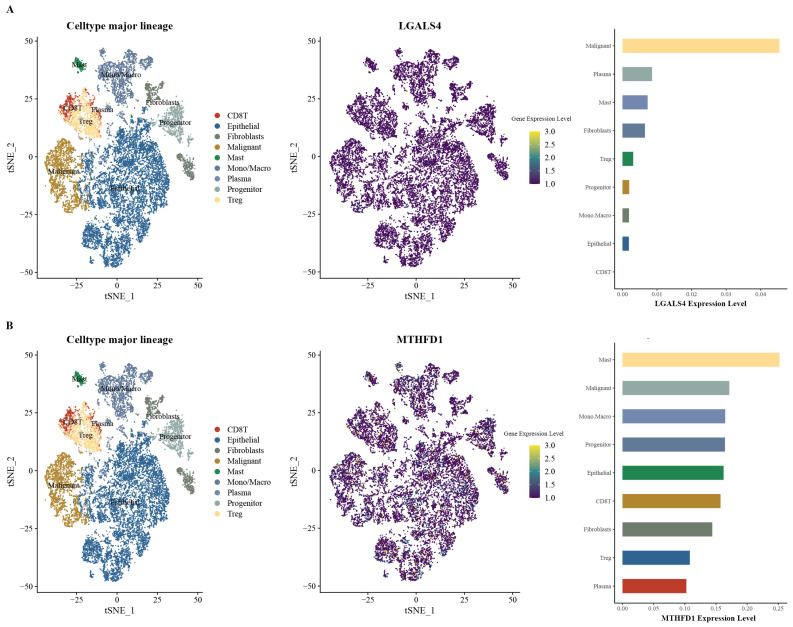
The single-cell expression of LGALS4 and MTHFD1 in PRAD: (**A**) the left is the t-SNE plot of single-cell clustering, where different colors represent different types of cells; the middle is the expression distribution of LGALS4 in different cells; and the right is the LGALS4 expression abundance in different cells. (**B**) the left is the t-SNE plot of single-cell clustering, where different colors represent different types of cells; the middle is the expression distribution of MTHFD1 in different cells; and the right is the MTHFD1 expression abundance in different cells.

**Table 1 biomedicines-13-00185-t001:** Candidate drugs predicted using DSigDB.

Term	*p* Value	Odds Ratio	Combined Score	Genes
etoposide BOSS	0.003	688.6	4000.6	LGALS4
1-(5-deoxypentofuranosyl)-5-fluoropyrimidine- 2,4(1h,3h)-dione CTD 00001171	0.005	399.0	2106.4	LGALS4
choline CTD 00005662	0.005	399.0	2106.4	MTHFD1
capecitabine CTD 00003557	0.006	362.6	1880.6	LGALS4
lactose BOSS	0.010	197.0	903.8	LGALS4
choline hydroxide BOSS	0.011	185.9	842.3	MTHFD1
dl-methionine BOSS	0.017	119.5	489.4	MTHFD1
folic acid BOSS	0.018	110.1	442.2	MTHFD1
methotrexate BOSS	0.018	109.5	439.1	MTHFD1
dihydroergocristine HL60 DOWN	0.019	103.7	410.4	MTHFD1
zinc sulfate CTD 00007264	0.028	68.9	245.2	MTHFD1
paclitaxel CTD 00007144	0.049	39.6	119.8	LGALS4
alprostadil HL60 DOWN	0.049	39.1	117.6	MTHFD1

## Data Availability

Data are provided within the manuscript or in the Appendix A.

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
