# Peer review of "Identifying MTHFD1 and LGALS4 as Potential Therapeutic Targets in Prostate Cancer Through Multi-Omics Mendelian Randomization Analysis"

_biomedicines, 2025, doi:10.3390/biomedicines13010185_

Round 1
Reviewer 1 Report
Comments and Suggestions for Authors
The manuscript outlines an approach that aids in identifying novel genes/proteins targeting prostate cancer. The theory and the study design merit acknowledgment. Nonetheless, subsequent experimental tests, such as H&E staining and PCR, might be helpful to ascertain the practicality and reliability of this method. Considering the originality and significance of this manuscript, a minor revision based on the following recommendations is advised.
1. Why is a paragraph about TME abruptly included in the introduction section? Please highlight the connection between this paragraph and the other paragraphs within the introduction section.
2. Figure 2. Consider including entire names in the caption. For example: UKB-PPP, SNP, and CI.
3. Figure 5. It is recommended to consolidate all genes and groups into a single figure, such as a volcano plot, to represent the mean expression levels of various genes.
4. Figure 6. Please prevent the overlap of lines and group designations. For the X-axis, please utilize "Time (months)" or a suitable temporal unit.
5. Figure 7. Instead of asterisks, the exact P value is recommended.
6. Figure 8. Are NK cells and B cells available for single-cell sequencing?
7. Figure 8. No noticeable differences among the various cell lines based on the mid figures, and all gene expression levels are visually below 1.5. What is the reason for the elevated gene expression in the malignant cell on the right?
8. Figure 8. Please eliminate the letter b from the middle figure. Clarify the x-axis value of the right figures. A violin plot illustrating gene expression levels and standard deviations is recommended.
Author Response
Response to Reviewer Comments
Dear Reviewers:
Thank you for your letter and for the reviewers’ comments concerning our manuscript entitled “Identifying MTHFD1 and LGALS4 as potential therapeutic targets in prostate cancer through multi-omics Mendelian ran-domization analysis’’ (ID: biomedicines-3407735). These comments are all valuable and very helpful for revising and improving our paper, as well as the important guiding significance to our researches. We have studied comments carefully and have made correction which we hope meet with approval.
Comments 1: Why is a paragraph about TME abruptly included in the introduction section? Please highlight the connection between this paragraph and the other paragraphs within the introduction section.
Response 1: We thank the reviewer for pointing this out. We included a brief section on the tumor microenvironment (TME) in the introduction to highlight its relevance to the identification of prostate cancer-related genes. To improve the flow and connection with other sections, we will revise the manuscript to add a clearer transitional sentence linking the TME section with the rest of the introduction.
Comments 2: Figure 2. Consider including entire names in the caption. For example: UKB-PPP, SNP, and CI.
Response2: We agree with the reviewer’s suggestion and will update the caption of Figure 2 to include the full names of terms such as UKB-PPP, SNP, and CI, along with definitions where necessary, to ensure clarity for the readers.
Comments 3: Figure 5. It is recommended to consolidate all genes and groups into a single figure, such as a volcano plot, to represent the mean expression levels of various genes.
Response 3: We appreciate the reviewer’s recommendation. We will combine all genes and groups into a single volcano plot to represent the mean expression levels of various genes more effectively. This will allow a clearer visualization of the data.
Comments 4: Figure 6. Please prevent the overlap of lines and group designations. For the X-axis, please utilize "Time (months)" or a suitable temporal unit.
Response 4: We will revise Figure 6 to prevent any overlap between the lines and group labels. Additionally, we will change the X-axis label to "Months", as suggested, to provide better clarity on the temporal aspect of the data.
Comments 5: Figure 7. Instead of asterisks, the exact P value is recommended.
Response 5: We will replace the asterisks in Figure 7 with the exact p-values to provide more precise information and ensure that the statistical significance is clear.
Comments 6: Figure 8. Are NK cells and B cells available for single-cell sequencing?
Response 6: We appreciate the reviewer's inquiry regarding the availability of NK cells and B cells for single-cell sequencing. Based on the data we used in our study, these specific cell types were not prominently identified or analyzed in this dataset (GSE176031). This might be due to the inherent limitations of this dataset or the tissue source used for sequencing. However, single-cell RNA sequencing is indeed capable of identifying NK cells and B cells in appropriate datasets, provided these populations are adequately represented in the sample.
Comments 7: Figure 8. No noticeable differences among the various cell lines based on the mid figures, and all gene expression levels are visually below 1.5. What is the reason for the elevated gene expression in the malignant cell on the right?
Response 7: Thank you for your insightful observation regarding the elevated gene expression in the malignant cells on the right-hand side of Figure 8. Although the expression levels of LGALS4 and MTHFD1 in prostate cancer cells are relatively low (all below 1.5, as indicated by the dark blue shading in the t-SNE plots), their average expression levels in malignant cells are indeed slightly higher compared to other cell types. Specifically, for both genes, the mean expression levels in malignant cells remain below 1.0 but are still elevated relative to non-malignant populations. This reflects their biological roles in processes such as tumor progression and metabolic adaptation, which are more active in malignant cells. To address this, we will update the figure to include more detailed annotations and a clearer visual distinction for mean expression levels. Additionally, we will revise the figure legend to explain these trends explicitly, making it easier for readers to interpret the data and understand the biological significance of these findings. Thank you for highlighting this point, which will improve the clarity and impact of our presentation.
Comments 8: Figure 8. Please eliminate the letter b from the middle figure. Clarify the x-axis value of the right figures. A violin plot illustrating gene expression levels and standard deviations is recommended.
Response 8: We agree with the reviewer’s suggestions to improve clarity. We will remove the letter "b" from the middle figure to ensure consistency and avoid confusion. For the x-axis in the right-side figures, we will add a clearer label, "Gene Expression Level", to explicitly define the variable being represented. While we understand the suggestion to replace the current plots with violin plots, the current bar plots are more suitable for conveying mean expression levels across cell types in this specific context. Violin plots are generally used for displaying the distribution and variability of data points, but here the mean expression across cell types is the primary focus. We are grateful for these constructive comments and will revise the manuscript accordingly to address these points. Thank you again for your valuable feedback.
Reviewer 2 Report
Comments and Suggestions for Authors
Comments to the Authors
Prostate cancer, a leading cause of cancer deaths in men worldwide, is treated with surgery, radiotherapy, and hormone therapy, but improving outcomes relies on advancing drug target screening for better diagnosis and treatment. The authors Huan Han et al. present a work in which they identified MTHFD1 and LGALS4 as potential preventive targets for Prostate adenocarcinoma (PRAD).
There are no significant technical or conceptual concerns in the manuscript that would hinder its publication.
Nonetheless, I suggest the following edits to improve its clarity and make it more accessible to a wider audience:
1. eQTLGen, UK Biobank, and deCODE are European databases, and most data are obtained from European populations. This limits the generalizability of the results to other populations and may lead to population-specific biases. It is unclear why the authors chose these three European databases. Can the authors justify their choice?
2. Please give full designation of some abbreviations: PRAD (line 16 ), HEIDI (line 15), PARD (line 275), PENT (line 272), PTEN (line 273).
3. No discussion of how these findings integrate with existing clinical strategies, particularly concerning resistance mechanisms to androgen deprivation therapy (ADT).
Author Response
Response to Reviewer Comments
Dear Reviewers:
Thank you for your letter and for the reviewers’ comments concerning our manuscript entitled “Identifying MTHFD1 and LGALS4 as potential therapeutic targets in prostate cancer through multi-omics Mendelian randomization analysis” (ID: biomedicines-3407735).These comments are all valuable and very helpful for revising and improving our paper, as well as the important guiding significance to our researches. We have studied comments carefully and have made correction which we hope meet with approval.
Comments 1: eQTLGen, UK Biobank, and deCODE are European databases, and most data are obtained from European populations. This limits the generalizability of the results to other populations and may lead to population-specific biases. It is unclear why the authors chose these three European databases. Can the authors justify their choice?
Response1: Thank you for your valuable comment. We appreciate your concern regarding the use of European databases and the potential limitations in generalizing our findings to other populations.
Regarding the choice of eQTLGen, UK Biobank, and deCODE, we selected these databases for the following reasons:
Firstly, for studies involving Mendelian Randomization (MR), it is essential to use databases that consist of a homogeneous population to minimize confounding variables. MR relies on genetic variants as instrumental variables, and using data from a single population (in this case, European) helps reduce potential biases that could arise from population structure differences. Given that MR studies are sensitive to population stratification, focusing on a population with relatively homogenous genetic backgrounds ensures more accurate and reliable results. Secondly, the three databases we selected (eQTLGen, UK Biobank, and deCODE) are well-established, comprehensive resources with high-quality data on genetic variants, gene expression, and associated phenotypes. They provide a large sample size, which enhances statistical power and allows for more robust analyses of gene-environment interactions. These resources have been widely used in genetic epidemiology and have been extensively validated in the context of various traits and diseases. Finally, although these databases primarily consist of European populations, they have been extensively studied in the context of gene expression, single-nucleotide polymorphisms (SNPs), and complex diseases, including cancer. Our research aims to investigate the genetic basis of prostate cancer (PCa) susceptibility and treatment outcomes, and these databases offer relevant genetic and phenotypic data that are aligned with our research objectives.
We acknowledge the limitations of using predominantly European data and understand that this may affect the generalizability of our findings to other populations. However, due to the quality, depth, and relevance of the available data in these European cohorts, we believe they represent the most suitable resources for our study at this stage. Future work may explore more diverse datasets to validate our findings across different populations. We hope this clarification addresses your concern. Thank you again for your thoughtful feedback.
Comments 2: Please give full designation of some abbreviations: PRAD (line 16), HEIDI (line 15), PARD (line 275), PENT (line 272), PTEN (line 273).
Response 2: We are very sorry for our negligence. The full designation of abbreviations has been added and marked in red in the revised paper. We appreciate for reviewers’ warm work eastly and hope that the correction will meet with approval.
Comments 3: No discussion of how these findings integrate with existing clinical strategies, particularly concerning resistance mechanisms to androgen deprivation therapy (ADT).
Response 3: Thank you for your valuable comment. We understand the importance of discussing how our findings integrate with existing clinical strategies, especially regarding resistance mechanisms in prostate cancer (PCa) treatment. We have revised our discussion to address this point.